# The Mechanisms of Different Light Supply Patterns in the Nutrient Uptake and Chlorophyll Fluorescence of Hydroponic Lettuce

**Yanwei Liu** [1,2,3] , **Ni Cao** [1,2,3], **Xiaolan Shi** [4], **Fei Meng** [1,2,3], **Yingjie Zhou** [1,2,3], **Haidong Wang** [1,2,3,*] **and Qiliang Yang** [1,2,3,*]

[1] Faculty of Modern Agricultural Engineering, Kunming University of Science and Technology, Kunming 650500, China; 20110170@kust.edu.cn (Y.L.); 20212214006@stu.kust.edu.cn (N.C.); 20212214007@stu.kust.edu.cn (F.M.); zhouyingjie@kust.edu.cn (Y.Z.)

[2] Yunnan Provincial Field Scientific Observation and Research Station on Water-Soil-Crop System in Seasonal Arid Region, Kunming University of Science and Technology, Kunming 650500, China

[3] Yunnan Provincial Key Laboratory of High-Efficiency Water Use and Green Production of Characteristic Crops in Universities, Kunming University of Science and Technology, Kunming 650500, China

[4] College of Ecology and Environment, Southwest Forestry University, Kunming 650224, China; 20222214040@stu.kust.edu.cn

[*] Correspondence: wanghd@nwsuaf.edu.cn (H.W.); 20090042@kust.edu.cn (Q.Y.)

**Abstract:** Vertical agriculture has developed rapidly in recent years, pushing artificial light planting to new heights. Under indoor artificial light, the light supply mode has practical significance in studying the impact of lighting conditions on plants. This experiment involved five intermittent light supply modes (with a light period of 16 h and a dark period of 8 h (16/8) as the control group, with a light period of 8 h and a dark period of 4 h repeated twice a day (8/4), a light period of 4 h and a dark period of 2 h repeated four times a day (4/2), a light period of 2 h and a dark period of 1 h repeated eight times a day (2/1), and a light period of 1 h and a dark period of 0.5 h repeated 16 times a day (1/0.5) (as the experimental groups). A total of ten treatments were combined before the continuous light supply mode (B) and after the continuous light supply mode (A). Through experimental settings, we explored the response mechanisms to intermittent and continuous light supply modes as they pertain to lettuce growth, nutrient content, photosynthetic parameters, and light stress. Through research, it was found that continuous light supply significantly increased plant height, root length, aboveground dry and fresh weight, and the underground dry and fresh weight of lettuce. The treatments with a light period 4 h/dark period 2 h (4/2) and a light period 8 h/dark period 4 h (8/4) significantly increased the N, P, K, and Cu contents. Additionally, continuous light supply helped stabilize the Mg, Ca, and Mn contents across all treatment groups. Stomatal closure has been found to cause a decrease in the rate of net photosynthesis, transpiration, and intercellular $CO_2$ concentration. The energy absorbed by antenna pigments is significantly increased when participating in photochemical reactions; however, continuous light supply has been observed to reduce the absorption flux per reaction center (ABS/RC), trapped energy flux per reaction center ($TR_0$/RC), electron transport flux per reaction center ($ET_0$/RC), and the probability that a trapped exciton moves an electron into the electron transport chain beyond $QA^-$ (at t = 0) ($ET_0/TR_0$). Conversely, the electron transport flux per cross section ($ET_0$/CS) has been found to increase significantly. In summary, among the intermittent light supply treatment groups, the 2/1 treatment group showed the best response to growth indicators, nutrient absorption, and photosynthesis, and could improve the quality of lettuce without adding additional light energy. Continuous light supply in the short term can improve the growth and nutrient absorption of lettuce; both of the two light supply modes produced light stress on lettuce, and this light stress caused by non-circadian rhythm forced the lettuce to increase its photochemical quenching (qI) and electro transport flux crossover ($ET_0$/CS). This paper may provide a theoretical reference for the use of light supply modes in plant factories to improve vegetable yield, and for the study of the response mechanism of light stress under non-circadian rhythm.

**Keywords:** chlorophyll fluorescence; lettuce; light supply mode; nutrient absorption; plant factory

## 1. Introduction

### 1.1. The Application of Light Supply Modes in Plant Factories

Plant factories are efficient agricultural systems that achieve the annual planned production of crops in vertical three-dimensional spaces through high-precision environmental control under completely enclosed or semi-enclosed conditions. In recent years, lettuce cultivation has gradually relied on plant factories to produce high quality and high volume [1]. In plant factories, light regulation is an important environmental factor affecting plant growth. The potential of actively implementing dynamic lighting strategies to control plant growth and development has excellent prospects for plant cultivation in the future [2]. Light-emitting diodes (LEDs) have the advantages of high energy efficiency, low cost, and long lifespan. Therefore, they are often used as artificial light sources in plant factories for supplementary lighting [3]. We can easily accomplish the regulation of the photoperiod in plant factories. Sometimes, lighting conditions that rarely or never occur in nature are also applied to increase crop yield and improve quality, such as continuous light supply at night and multiple light–dark period exchanges within a day. Panjai et al. [4] pointed out that the content of lycopene and some secondary metabolites can be significantly increased via continuous red light irradiation after tomato picking. Dong et al. [5] proposed that wheat growth and physiological indicators are more sensitive to intermittent light during the heading and flowering stages. Simultaneously testing whether establishing static signal state light can be used as a growth regulator under continuous light can eliminate the powerful impact of photoperiodic pathways [6]. The future of utilizing light supply modes to manipulate plant habits and productivity is full of opportunities.

### 1.2. The Characteristics of Light Supply Modes

Light affects plant photosynthesis and light morphogenesis [7]. Light affects the growth and development of plants through three main aspects: light intensity, light quality, and the light cycle [8]. Research on light intensity mainly focuses on setting different light gradients through the response of indicators to find the optimal light conditions for plant growth and storage [9,10]. The research on light quality mainly focuses on two aspects. The first aspect is the study of the effects of different wavelengths of light on plants [11], such as ultraviolet light (UV-A, UV-B) [12], far-red light [13], and green light [14]. The second aspect focuses on finding the optimal ratio of light for plant growth [15,16]. Previous studies on light intensity and quality have become more mature. However, research on light supply modes in the photoperiod has yet to be in-depth.

Light supply modes can be divided into three types: continuous light supply, alternating light supply, and intermittent light supply.

In their research on alternating light supply, Ohtake et al. [17] pointed out that continuous exposure to alternating red and blue light can promote the growth of lettuce while maintaining its nutritional content. Kuno et al. [18] pointed out that alternating irradiation produced better results in response to indicators compared to simultaneous exposure to red and blue light. Under the same energy consumption, lettuce with red and blue light supply exchanged once in 8 h and 1 h had a higher yield, and lettuce with red and blue light supply exchanged once in 4 h and 2 h had higher nutritional value [19]. Meanwhile, the alternating mode can provide a method for in-depth research on the relationship between light quality and plants. However, the lighting control technology for alternating light supply is relatively cumbersome, and the equipment cost is high.

In the study of continuous light supply, continuous red light irradiation on tomatoes after picking was found to significantly increase their content of lycopene and some secondary metabolites [4]. However, tomatoes grown under continuous light can exhibit potential fatal spot yellowing, which counters increasing tomato yield [20], and Matsuda

pointed out that the light quality and intensity of overnight LED irradiation can affect the growth and damage level of tomato seedlings [21]. Zha stated in his paper that 15 days of continuous light could significantly improve the quality of lettuce, while 30 days of continuous light had no positive impact on lettuce yield [22,23]. Based on the above viewpoints, we can conclude that long-term light supply has no positive effect on lettuce growth, and consumes much energy, but short-term continuous light supply can improve plant yield.

As mentioned by Avgoustaki [24], research on intermittent light supply determines that intermittent light can effectively reduce indoor light consumption by providing an optimal light cycle of 16 h and an intermittent light supply of 14 h for basil. Based on the stability of basil's growth state, it is determined that intermittent light can effectively reduce indoor light supply energy consumption without adverse effects on growth rate and biomass. Chen pointed out in her paper that under 16 h of daily light conditions, all treatments significantly increased the glucose content in lettuce, and some of them also improved the taste of lettuce [25]. Compared with continuous light supply, intermittent light supply increases biomass yield, and over time, intermittent light supply is more suitable for biomass accumulation [26]. Intermittent light supply can improve plant growth and physiological indicators without increasing light energy consumption. This can be seen as a way to save energy.

In the existing literature, research on light supply modes mainly focuses on plant growth indicators and metabolites. The absorption of nutrients under light supply modes, whether light stress occurs, and the response mechanism of plants to stress still need to be clarified.

### 1.3. The Effects of Light Supply Mode on Plants

In previous studies, light supply patterns can impact plant yield, quality, and photosynthesis. Liu et al. [27] studied the effect of light quality on the dry matter accumulation, nutrient element content, and cumulant of lettuce under three nitrogen supply levels under LED red and blue light continuous illumination before harvest. The research shows that continuous light supply before harvest can promote the accumulation of lettuce's dry matter, and increase the Ca and Mg content. Shao et al. [28] studied the effects of different light rhythms before harvest on the growth of hydroponic lettuce and the absorption of nutrient elements. The study showed that the light rhythm of 8/4 h (T12) significantly increased the dry and fresh weight of lettuce, and expanded the leaf area; under T12 treatment, the cumulant of N, C, P, K, Ca, Mg elements was the highest. When studying the photosynthetic characteristics of different types of lettuce under continuous light supply, Cha et al. [29] found that continuous light supply led to varying degrees of decrease in the net photosynthetic rate, stomatal conductance, intercellular $CO_2$ concentration, and transpiration rate of lettuce.

In the study of light quality and light intensity, the light source provided by continuous light supply and intermittent light supply is different from the circadian rhythm of plant growth in natural light, so it is necessary to consider whether this non-circadian rhythm light supply causes any stress on plant growth. We know that any form of stress can ultimately be attributed to antioxidant stress [30]. Chlorophyll fluorescence is a vital response indicator in describing stress. The analysis of chlorophyll fluorescence provides detailed information about the state and function of photosystem II PS (II) reaction center, light capture antenna complex, and PS II donor side and receptor side [31]. Hoffmann et al. [32] suggested that blue light can serve as a light signal and stress factor, enhancing the accumulation of antioxidant substances in plant species under intermittent exposure to high/low blue light. Lavaud et al. [33] showed that the transformation of diadinoxanthin (DD) to diatoxanthin (DT) rapidly leads to a very strong quenching of the chlorophyll fluorescence yield under high light intensity. Previous studies on light stress have mainly focused on the effects of high light, low light, and monochromatic radiation light on plants. The effects of light supply modes on light stress have yet to be deeply explored.

*1.4. The Research Purpose of This Article*

Through the above analysis, this article aims to investigate the effects of intermittent light supply on lettuce growth, nutrient absorption, and photosynthesis, in order to find the optimal intermittent light supply treatment for lettuce. We conducted a short-term (72 h) continuous light supply treatment on lettuce in the pre-harvest period to explore the impact of short-term continuous light supply on lettuce; by explaining the effect of light stress on the chlorophyll fluorescence of lettuce under non-circadian rhythm, we hope to understand the response mechanism of lettuce to stress.

## 2. Materials and Methods

*2.1. Experimental Site*

The experiment was conducted in the fully artificial light plant factory of Kunming University of Science and Technology (24°50′45″ N, 102°51′55″ E). The experiment started on 10 June 2022 and ended on 20 October 2022.

*2.2. Plant Materials, Hydroponic Setup and Growing Conditions*

The experiment used lettuce (*Lactuca sativa* L. cv. 'Yidali') as the experimental material. We soaked the seeds in clean water for 3–5 h, removed and rinsed them with water, then wrapped them in gauze, squeezed out the water slightly, and placed them in a well-ventilated place. They were washed once a day. After exposure, we sowed the seeds on a 2.5 cm square sponge block, with one seed per block. In the beginning, we started by watering with clean water, then watering with nutrient solution after the leaves were flattened. After 5 days of hydroponic growth in an environmentally controllable growth chamber, the seedlings were planted in hydroponic chambers with different light environments for cultivation. Each hydroponic incubator (60 cm × 80 cm × 10 cm) contained 24 plants spaced 14 cm apart. Air temperature, relative humidity and $CO_2$ level were, respectively, maintained at 24/20 °C (day/night), 60% and 900 μmol mol$^{-1}$. All lettuce seedlings grew in the same conditions during the seedling cultivation period.

*2.3. Experimental Design and Treatment*

The experimental setup was supplemented by red blue light with a light quality of 5:1 and white light. The light intensity of red and blue light is 150 μmol m$^{-2}$ s$^{-1}$, and the light intensity of white light is 50 μmol m$^{-2}$ s$^{-1}$. The total light intensity was controlled to 200 μmol m$^{-2}$ s$^{-1}$. Five intermittent light supply treatments were set up in the experiment, with a light period of 16 h and a dark period of 8 h (16/8) as the control group, and a light period of 8 h and a dark period of 4 h repeated twice a day (8/4), a light period of 4 h and a dark period of 2 h repeated four times a day (4/2), a light period of 2 h and a dark period of 1 h repeated eight times a day (2/1), and a light period of 1 h and a dark period of 0.5 h repeated 16 times a day (1/0.5) as the experimental groups. Based on the five treatments of intermittent light supply, the experiment featured a short-term (72 h) continuous light supply in the pre-harvest period. For comparison before and after continuous light supply, two treatments were set up: before continuous light supply (B), and after continuous light supply (A). There were a total of ten treatments. The experimental treatment is shown in Table 1.

**Table 1.** Experimental light treatment.

|  | 1/0.5 | 2/1 | 4/2 | 8/4 | 16/8 |
|---|---|---|---|---|---|
| Before continuous light supply (B) | 1/0.5 B | 2/1 B | 4/2 B | 8/4 B | 16/8 B |
| After continuous light supply (A) | 1/0.5 A | 2/1 A | 4/2 A | 8/4 A | 16/8 A |

### 2.4. Nutrient Solution Preparation and Management

We used an improved Hoagland solution for all lettuce plants [34]. The nutrient solution was renewed per week, which electrical conductivity adjusted to 1.2–1.3 ms cm$^{-1}$ and pH maintained at 5.8–6.0.

### 2.5. Growth and Yield of Lettuce

After 40 days of transplanting, three lettuce plants were randomly selected from each experimental group, wiped clean with water, and placed on a balance to measure the fresh weight of the aboveground and underground parts (accurate to 0.001 g). In an oven, they were blanched at 105 °C for 15 min, then dried at 80 °C until a constant weight was recorded for the dry weight of the lettuce above and below ground. A vernier caliper and a tape measure were used to measure the root and leaf lengths of lettuce plants. Finally, the root to shoot ratio (aboveground dry weight/underground dry weight) was calculated.

### 2.6. Nutrient Contents of Lettuce
#### 2.6.1. C, N, P, K Content

Some 20–30 mg of plant samples that have been ground and dried through a 0.25 mm sieve were weighed and poured into a dry hard test tube. Slowly, 10 mL of 0.4 mol/L potassium dichromate solution and 10 mL of concentrated sulfuric acid were added. The test tube was placed in a paraffin oil bath at a temperature of 185–195 °C, allowing the liquid level inside the test tube to immerse below the paraffin level. The paraffin oil bath was adjusted to 170–180 °C, and timing started when the solution in the tube slightly boiled; the test tube was removed after five minutes. After slightly cooling, all the digestion solution in the test tube was transferred into a 150 mL triangular flask and washed with washing solution until the volume reached 60–70 mL to maintain a concentrated sulfuric acid concentration of 1–1.5 mol/L. Then, 3–4 drops of o-phenanthroline indicator were added, and the mixture was titrated with a 0.2 mol/L standard ferrous sulfate solution. The endpoint was deemed to be when the solution changed from yellow, green, and grayish blue to brownish red. We recorded the amount of iron used. Two blank experiments were performed while measuring the sample, and the average value was taken to obtain the C content.

The N, P, and K content were measured using the $H_2SO_4$-$H_2O_2$ digestion method. The digestion treatment proceeded as follows. Some 0.5 g of lettuce stems and leaves were weighed and transferred to the bottom of the digestion tube, and concentrated $H_2SO_4$ was added. The tube was shaken well overnight and heated in a low-heat digestion furnace; we waited for $H_2SO_4$ to emit white smoke, and then increased the temperature. When the solution showed a uniform brown-black color, we removed it. After slightly cooling, we added 10 drops of $H_2O_2$, then heated until slightly boiling and simmered for about 7–10 min. After slightly cooling, we repeated adding $H_2O_2$ and simmered again. We repeated this process until the digestion solution was colorless or transparent, and then heated for about 10 min to remove the remaining $H_2O_2$. After removal and cooling, the boiling solution was transferred into a 100 mL volumetric flask without damage and cooled to room temperature; the volume was then fixed. A blank test digestion solution was set up to correct errors in reagents and methods.

According to the principles and technology of plant physiological and biochemical experiments [35], 10 mL of digested solution after constant volume was absorbed, and the nitrogen content was measured using the Kjeldahl method. After the plant sample was subjected to Kjeldahl digestion and at a constant volume, a portion of the digestion solution was extracted and alkalized to convert the ammonium salt into ammonia. After distillation, it was absorbed with $H_3BO_3$. The ammonia absorbed in boric acid can be directly titrated with standard acid, and the mixed indicator of methyl red bromocresol green determines the endpoint.

Some 20 mL of digested solution was sucked into a 50 mL volumetric flask, and two drops of dinitrophenol indicator were added; 6 mol/L NaOH was added to neutralize

until just yellow. Then, 10 mL of ammonium vanadate molybdate reagent was added, and the volume was fixed with water. After 15 min, the P content was determined using a UV visible spectrophotometer (721G-100, Shanghai Instrument and Electronics Analytical Technology Co., Ltd., Shanghai, China) at 440 nm, and the blank test digestion solution was used as the blank control group.

Then, 10 mL of digested solution was put into a volumetric flask. After constant volume with water, we directly measured the K content with a flame photometer (FP6431, Shanghai Instrument and Electronics Analytical Technology Co., Ltd., Shanghai, China).

### 2.6.2. Ca, Mg, Fe, Mn, Zn, Cu Content

The sample digestion method is the same as above. The prepared standard solution was injected into the inductively coupled plasma emission spectrometer to measure the intensity signal response value of the analytical spectral line of the element to be measured. A standard curve was drawn, with the concentration of the element to be measured as the abscissa and the intensity response value of the analytical spectral line as the ordinate. The blank solution and sample solution were injected into the inductively coupled plasma emission spectrometer, respectively; we then measured the signal response value of the spectral line intensity of the element to be measured, and obtained the concentration of the element to be measured in the digestion solution according to the standard curve.

The calculation formula for the content of Ca, Mg, Fe, Mn, Zn and Cu in the sample is as follows:

$$X = ((\rho - \rho_0) \times V \times f)/m \tag{1}$$

In the formula, X represents the content of the element to be tested in the sample (mg/kg), $\rho$ represents the mass concentration of the measured elements in the solution (mg/L), $\rho_0$ represents the mass concentration of the measured elements in the blank solution (mg/L), V represents the constant volume of sample digestion solution (mL), f represents sample dilution ratio, and M represents Sample mass (g).

### 2.7. Photosynthetic Pigment Measurement

We took fresh plant leaves, removed their midrib, and cut them into small pieces. We weighed 2 g of a freshly cut sample, put it into a mortar, added a small amount of quartz sand and calcium carbonate to 3 mL of 95% ethanol, ground it into a homogenate, added 10 mL of ethanol, and continued to grind until the tissue turned white. After centrifugation in a centrifuge, the supernatant was filtered using filter paper moistened with ethanol. Finally, it was diluted to 100 mL with ethanol and shaken well. The absorbance of photosynthetic pigments was measured using a UV spectrophotometer (721G-100, Shanghai Instrument and Electronics Analytical Technology Co., Ltd., Shanghai, China) at wavelengths of 665 nm, 645 nm, and 470 nm. Each group was measured three times, and the average value was obtained.

### 2.8. Photosynthetic Parameters

Lettuce plants were selected randomly during processing, and the fourth fully unfolded leaf was measured. The net photosynthetic rate (Pn), stomatal conductance (Gs), transpiration rate (Tr), and intercellular $CO_2$ concentration (Ci) were measured using a portable photosynthetic instrument (LI-6400XT, LI-COR, Lincoln, NE, USA). Each group was measured six times, and the average value was obtained.

### 2.9. Chlorophyll Fluorescence Parameters

Using a chlorophyll fluorescence meter (OS5p+, Opti-Sciences, Inc., Huston, NH, USA), we measured the chlorophyll fluorescence of the lettuce leaves. After 40 days of transplanting, lettuce plants were randomly selected from each treatment, and a third leaf was taken for chlorophyll fluorescence measurement. Before testing, the leaves were placed in leaf clips, avoiding the position of leaf veins, and dark adaptation was performed for

20 min before measurement. Table 2 lists the chlorophyll fluorescence parameters to be used in this article. Each group was measured six times, and the average value was obtained.

**Table 2.** Significance of chlorophyll fluorescence-related parameters.

| Formulae and Terms | Illustrations |
|---|---|
| $F_0$ | Minimal recorded fluorescence intensity |
| $Fm$ | Maximal recorded fluorescence intensity |
| $Fv/Fm = (Fm - F_0)/Fm$ | Maximum quantum efficiency of PSII photochemistry |
| $\varphi P_0 = TR_0/ABS$ | Maximum quantum yield for primary photochemistry |
| $\Psi_0 = ET_0/TR_0$ | Probability that a trapped exciton moves an electron into the electro transport chain beyond $Q_{A^-}$ (at t = 0) |
| $Y(II) = Fv'/Fm'$ | Maximum efficiency of PSII |
| $Y(NPQ) = F/Fm' - F/Fm$ | Heat energy dissipated by a photoprotective mechanism |
| $Y(NO) = F/Fm$ | Passive dissipation of heat and fluorescing energy |
| $PIABS \equiv (RC/ABS)[\varphi P_0/(1 - \varphi P_0)][\psi_0/(1 - \psi_0)]$ | Performance index on an absorption basis |
| $PICS \equiv (RC/CS_0)[\varphi P_0/(1 - \varphi P_0)][\psi_0/(1 - \psi_0)]$ | Performance index on a cross-section basis (at t = 0) |
| $ABS/RC$ | Absorption flux per RC (reaction center) |
| $TR_0/RC$ | Trapped energy flux per RC (at t = 0) |
| $ET_0/RC$ | Electron transport flux per RC (at t = 0) |
| $DI_0/CS$ | Dissipated energy flux per CS (cross-section) (at t = 0) |
| $ET_0/CS$ | Electron transport flux per CS (at t = 0) |
| $RC/CS$ | Density of reaction centers |

*2.10. Data Processing and Analysis*

Excel 2016 was used for data statistics, SPSS 25.0 was used for data significance analysis, and Origin 2022 was used to draw graphs.

**3. Results**

*3.1. Growth Measurement*

The growth indicators of lettuce are shown in Table 3. In the treatment group for plant height, before continuous light supply, the plant heights of the 2/1 B, 8/4 B, and 16/8 B treatments were significantly higher than those of the 1/0.5 B and 4/2 B treatments, and the maximum value appeared in 2/1 B. After continuous light supply, the plant height of all treatments was significantly improved, with the 2/1 A treatment group significantly increasing by 12.9%, 10.7%, and 6.3% compared to 1/0.5 A, 4/2 A, and 16/8 A. Regarding root length, before continuous light supply, the 2/1 B and 4/2 B treatment groups had significantly higher root lengths than the other treatment groups, but there was no significant difference between the two. There was no significant difference between the treatments after continuous light supply, indicating that continuous light supply made the root growth trend of lettuce in the treatment tend to be consistent, and continuous light supply also significantly increased the root length of lettuce. Regarding the fresh aboveground weight of lettuce, both before and after continuous light supply, the 2/1 treatment group had significantly higher weights than the other treatment groups, indicating that intermittent light supply helped to improve the fresh aboveground weight, and the same trend was observed in the dry aboveground weight. For the underground fresh weight index, the 2/1 B treatment showed significantly higher values than the other treatments before continuous light supply, which is consistent with the fresh aboveground weight. However, after continuous light supply, the fresh underground weights of the 2/1 A, 4/2 A, and 16/8 A treatments were significantly heavier than others, indicating that continuous light supply increased the fresh underground weight of the treatment group, and the maximum value still appeared at 2/1 A. Regarding the underground dry weight, before and after continuous light supply, the 2/1 treatment showed significantly higher values than the other treatments. For the root to shoot ratio, the 8/4 B treatment had a significantly higher value before continuous light supply than the 2/1 B and 4/2 B treatments. After continuous light supply, there was no significant difference among the treatments.

**Table 3.** Effects of different light supply modes on the plant height, root length, underground/aboveground dry and fresh weight, and root to shoot ratio of lettuce.

| Treatment | Plant Height (cm) | Root Length (cm) | Overground Part (g) | | Underground Part (g) | | Root Shoot Ratio |
|---|---|---|---|---|---|---|---|
| | | | Fresh Weight | Dry Weight | Fresh Weight | Dry Weight | |
| 1/0.5 B | 13.92 ± 0.15 b | 25.60 ± 0.56 b | 43.59 ±2.36 b | 2.43 ± 0.23 b | 4.61 ± 0.20 c | 0.41 ± 0.02 b | 5.92 ± 0.19 ab |
| 2/1 B | 15.87 ± 0.49 a | 31.61 ± 0.33 a | 58.61 ±1.16 a | 4.13 ± 0.11 a | 9.59 ± 0.43 a | 0.95 ± 0.06 a | 4.36 ± 0.17 b |
| 4/2 B | 14.17 ± 0.43 b | 32.23 ± 1.17 a | 45.46 ±3.88 b | 2.50 ± 0.11 b | 6.15 ± 0.05 b | 0.47 ± 0.03 b | 5.33 ± 0.23 b |
| 8/4 B | 15.83 ± 0.38 a | 17.36 ± 0.63 c | 21.31 ±2.98 c | 1.07 ± 0.15 c | 1.62 ± 0.24 d | 0.14 ± 0.06 c | 8.36 ± 2.31 a |
| 16/8 B | 15.53 ± 0.55 a | 27.58 ± 2.49 b | 46.56 ±2.85 b | 2.78 ± 0.32 b | 6.06 ± 0.63 b | 0.44 ± 0.04 b | 6.41 ± 1.16 ab |
| 1/0.5 A | 16.08 ± 0.42 d | 32.59 ± 1.89 a | 44.38 ±2.26 c | 2.68 ± 0.03 c | 4.87 ± 0.13 b | 0.56 ± 0.04 c | 4.81 ± 0.25 a |
| 2/1 A | 18.15 ± 0.23 a | 34.14 ± 0.79 a | 77.81 ±1.67 a | 5.41 ± 0.16 a | 9.62 ± 1.07 a | 1.01 ± 0.12 a | 5.43 ± 0.48 a |
| 4/2 A | 16.39 ± 0.32 cd | 33.43 ± 0.60 a | 63.13 ±2.59 b | 3.46 ± 0.22 b | 8.85 ± 0.44 a | 0.72 ± 0.09 bc | 4.88 ± 0.87 a |
| 8/4 A | 17.44 ± 0.52 ab | 31.54 ± 1.75 a | 29.24 ±1.05 d | 1.70 ± 0.14 d | 3.83 ± 0.30 b | 0.26 ± 0.04 d | 6.69 ± 1.26 a |
| 16/8 A | 17.07 ± 0.16 bc | 34.35 ± 0.65 a | 67.76 ±2.51 b | 5.61 ± 0.38 a | 9.18 ± 0.94 a | 0.92 ± 0.14 ab | 6.17 ± 0.72 a |

Lowercase letters indicate the significant difference labeling of five intermittent light supply treatments, while uppercase letters indicate the significant difference labeling of the same intermittent light supply treatment before and after continuous light supply. There is significant difference in the same parameter with different letters at the level of $p < 0.05$. 1/0.5: Light period 1 h, dark period 0.5 h, repeated 16 times a day; 2/1: Light period 2 h, dark period 1 h, repeated 8 times a day; 4/2: Light period 4 h, and dark period 2 h repeated 4 times a day; 8/4: Light period 8 h, dark period 4 h, repeated twice a day; 16/8: Light period 16 h, dark period 8 h. B: Before continuous light supply; A: After continuous light supply. (Hereinafter the same).

### 3.2. C, N, P, K Content

Figure 1 shows the effect of different light supply modes on the content of C, N, P, and K. For the C content, there was no significant difference between the treatment groups before continuous light supply. After continuous light supply, the 1/0.5 A and 16/8 A treatment groups had significantly higher values than other treatments, but there was no significant difference between them. After continuous light supply, the C content in the 2/1 A and 4/2 A treatment groups decreased significantly compared to before continuous light supply, while the C content in the 16/8 A treatment group increased significantly compared to before continuous light supply. For N content, before continuous light supply, the 16/8 B treatment was significantly higher than other treatments, and the 4/2 B treatment was significantly lower. After continuous light supply, the N content of 4/2 A and 8/4 A treatments significantly increased, with 8/4 A and 16/8 A treatments significantly higher than 1/0.5 A and 2/1 A treatments, indicating that continuous light supply played a positive role in the accumulation of N content. For P content, the 16/8 B treatment group was significantly higher before continuous light supply than the other treatment groups. After continuous light supply, the P content of the 4/2 A treatment significantly increased, but the P content of the 2/1 A and 16/8 A treatments significantly decreased. Before continuous light supply, 1/0.5 B and 4/2 B were significantly higher for K content than the control group of 16/8 B. After continuous light supply, the contents of 4/2 A and 8/4 A were significantly increased, with 8/4 A being significantly higher than other treatment groups.

### 3.3. Mg, Ca, Fe, Mn, Zn, Cu Content

Figure 2 describes the effects of different light supply modes on Mg, Ca, Fe, Mn, Zn, and Cu content. Both intermittent and continuous light supply modes significantly impact the measured nutrient content, but have different responses to different nutrient contents. For the content of Mg, Ca, and Mn, before continuous light supply, the nutrient content showed a trend of increasing, then decreasing, and then increasing. Among them, for the contents of Mg and Ca, the 8/4 B treatment had significantly higher values than the other treatments. The 2/1 B treatment was significantly higher in Mn content than other treatments. After continuous light supply, the Mg, Ca, and Mn contents were relatively stable. For treatments 1/0.5 A, 4/2 A, and 16/8 A, after continuous light supply, the content of Mg, Ca, and Mn significantly increased, while the content of Mg, Ca, and Mn significantly decreased in treatments 2/1 A and 4/2 A. For Fe and Zn content, before continuous light

supply, the Fe and Zn contents of the 8/4 treatment group were significantly higher than those of the other treatment groups. However, there was no significant difference between the other treatments. After continuous light supply, the Fe content between the treatment groups showed a trend of increasing and then decreasing. Among them, the 1/0.5 A and 16/8 A treatments were significantly higher in Fe content before continuous light supply. In contrast, the other treatments were significantly lower in Fe content before continuous light supply. For Zn content, the Zn content of all treatment groups was lower than before continuous light supply, and there was also a trend of Zn content stabilizing between treatment groups. For Cu content, the Cu content of the 4/2 B and 8/4 B treatment groups was significantly lower than that of the other treatment groups. After continuous light supply, the Cu content showed an opposite trend, with the Cu content in the 4/2 A and 8/4 A treatment groups being significantly higher than that in the 1/0.5 A and 16/8 A treatment groups. The 4/2 B and 8/4 B treatment groups showed a significant increase compared to before continuous light supply, while the other treatment groups showed a significant decrease compared to before continuous light supply.

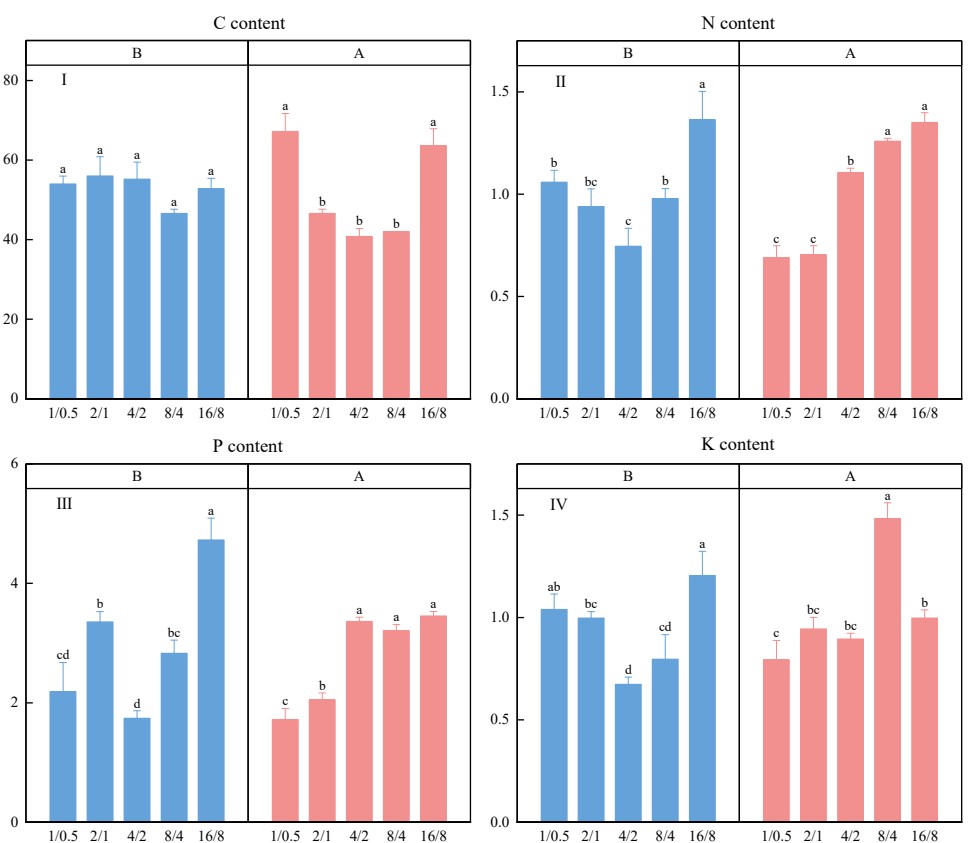

**Figure 1.** Histogram of the effect of different light supply modes on C, N, P, K content in plants. The bars represent the standard errors. There is significant difference in the same parameter with different letters at the level of *p* < 0.05. (**I**) C contents under intermittent light supply treatment before and after continuous light supply; (**II**) N contents under intermittent light supply treatment before and after continuous light supply; (**III**) P contents under intermittent light supply treatment before and after continuous light supply; (**IV**) K contents under intermittent light supply treatment before and after continuous light supply. B: Before continuous light supply; A: After continuous light supply.

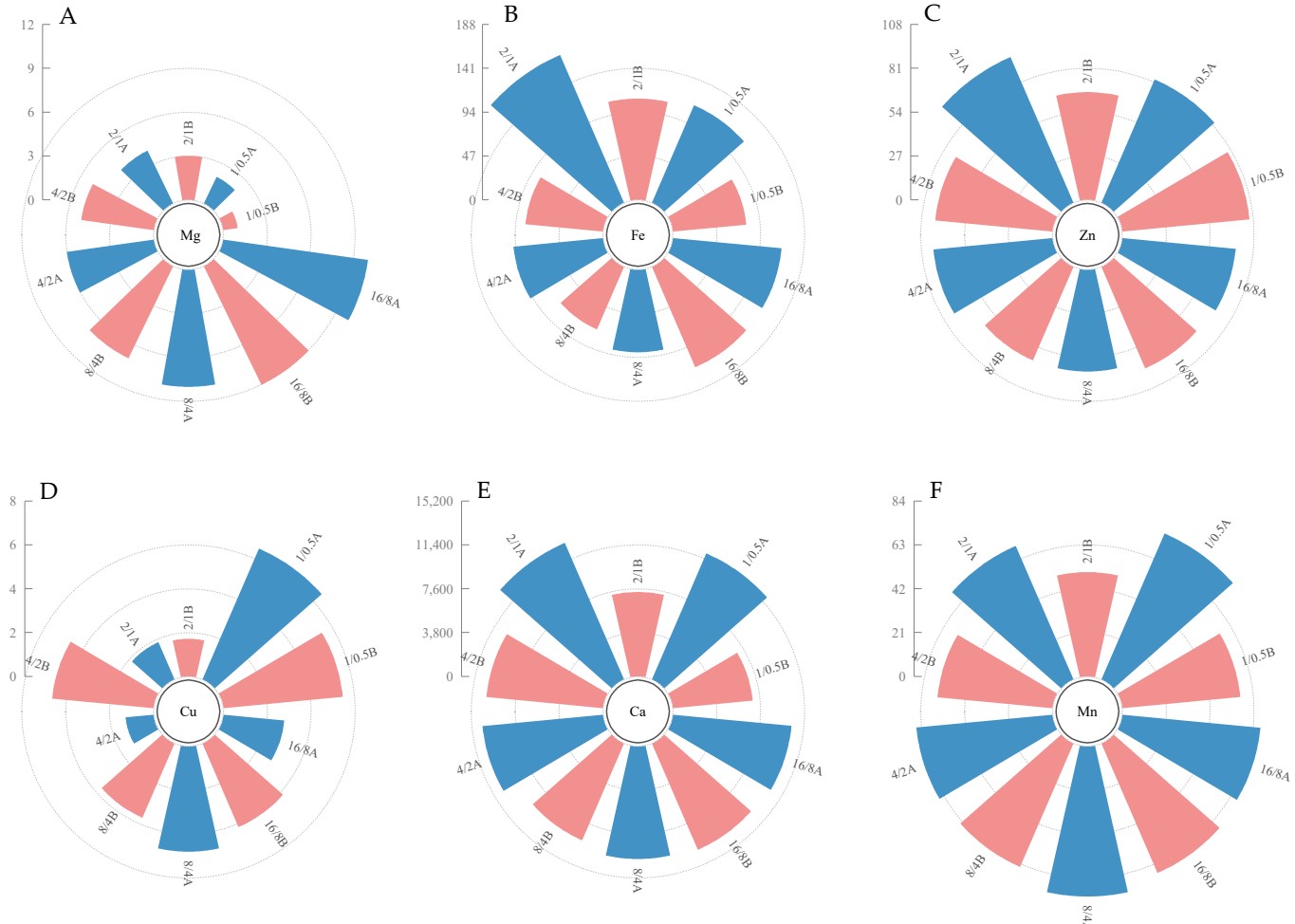

**Figure 2.** Histogram of effect of different light supply modes on Mg, Ca, Fe, Mn, Zn, Cu contents in plants. (**A–F**) Mg, Fe, Zn, Cu, Ca, and Mn contents under intermittent light supply treatment before and after continuous light supply. B: Before continuous light supply; A: After continuous light supply.

*3.4. Photosynthetic Pigment Measurement*

Figure 3 shows the effects of different light supply modes on chlorophyll a (Chla), chlorophyll b (Chlb), and carotene (Car). Continuous light supply significantly increased the chlorophyll content in the treatment group. Before continuous light supply, the chlorophyll content of the 4/2 B and 16/8 B treatments was significantly higher than that of the 2/1 B and 8/4 B treatments. After continuous light supply, the chlorophyll content of the 2/1 A treatment was the highest and significantly higher than that of the 1/0.5 A, 4/2 A, and 8/4 A treatments. For chlorophyll b, before continuous light supply, the chlorophyll b content in the 4/2 B treatment was the highest and significantly higher than that in the 2/1 B, 8/4 B, and 16/8 B treatments. However, there was no significant difference compared to the 1/0.5 B treatment. After continuous light supply, the chlorophyll b content in the 1/0.5 A treatment was the highest. However, the chlorophyll b content in the 4/2 treatment significantly decreased compared to before continuous light supply. There were no significant changes before and after continuous light supply in treatments 1/0.5, 2/1, and 16/8. For carotenoids, before continuous light supply, the 4/2 B treatment was significantly higher than the 2/1 B, 8/4 B, and 16/8 B treatments, but there was no significant difference compared to the 1/0.5 B treatment. After continuous light supply, the 4/2 A treatment was significantly lower than the 1/0.5 A, 2/1 A, and 16/8 A treatments. Except for the 16/8 A treatment, which showed a significant increase compared to before light supply, there was

no significant change in the other treatments. The control group showed good adaptability to light stress. In summary, under continuous light supply, the chlorophyll content was significantly increased, mainly in relation to the activity of the reaction center. The decrease in the chlorophyll b and carotene contents may be due to light damage to the photosystem, which hinders the transmission of light energy by antenna pigments.

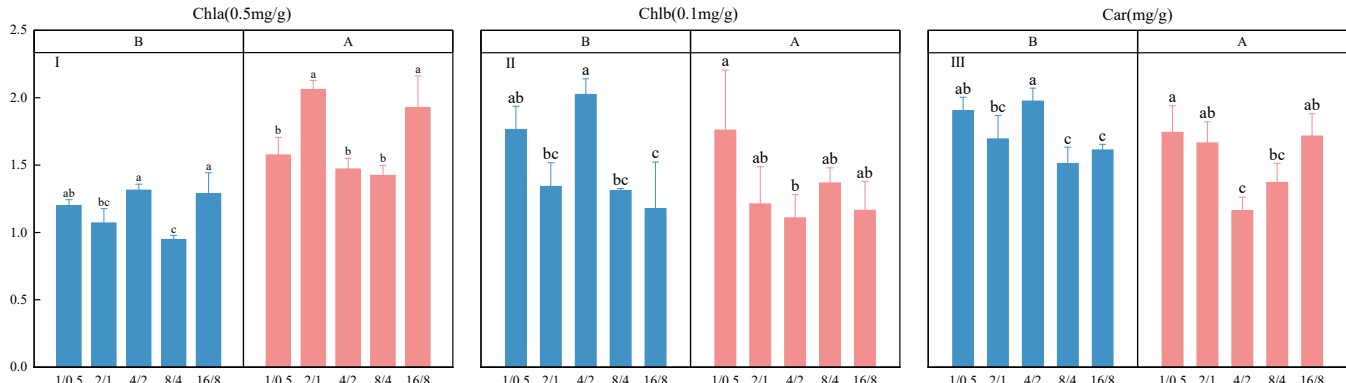

**Figure 3.** Bar of the effect of different light supply modes on chlorophyll a (Chla), chlorophyll b (Chlb) and carotene (Car) in Plants. The bars represent the standard errors. There is a significant difference in the same parameter with different letters at the level of $p < 0.05$. (**I–III**) Chla, Chlb and Car contents under intermittent light supply treatment before and after continuous light supply. B: Before continuous light supply; A: After continuous light supply.

### 3.5. Photosynthetic Parameters

Figure 4 shows the effects of different light supply modes on the net photosynthetic rate (Pn), stomatal conductance (Gs), transpiration rate (Tr), and intercellular $CO_2$ concentration (Ci). Pn reached its maximum value at 4/2 B before continuous light supply, followed by 2/1 B. Except for the 8/4 B treatment, the Pn of other intermittent light supply treatments was significantly higher than that of the control group 16/8 B, indicating that intermittent light supply treatment can significantly increase the net photosynthetic rate of lettuce, thereby increasing biomass accumulation. After continuous light supply treatment, Pn decreased to varying degrees. However, Pn increased in the 16/8 A treatment, indicating that the 16/8 A treatment group had better adaptability to light stress than others. For Gs, Tr, and Ci, the indicators decreased significantly after continuous light supply. This indicates that light stress significantly impacts Gs, Tr, and Ci. Before continuous light supply, Gs, Tr, and Ci showed a trend of increasing and decreasing, and their maximum values all appeared at 2/1 B. After continuous light supply, due to the damage of the light system, Gs and Ci showed no significant fluctuations. The 1/0.5 A, 2/1 A, and 4/2 A treatments in Tr showed better adaptability to light stress than the 8/4 A and 16/8 A treatment groups.

### 3.6. Chlorophyll Fluorescence Parameters

The changes in chlorophyll fluorescence in plants can to some extent reflect the impact of environmental factors on plants. By analyzing the rapid chlorophyll fluorescence induction kinetics curve under different environmental conditions, we can deeply explore the response mechanism of photosynthetic mechanisms, mainly PSII, to the environment under stress conditions.

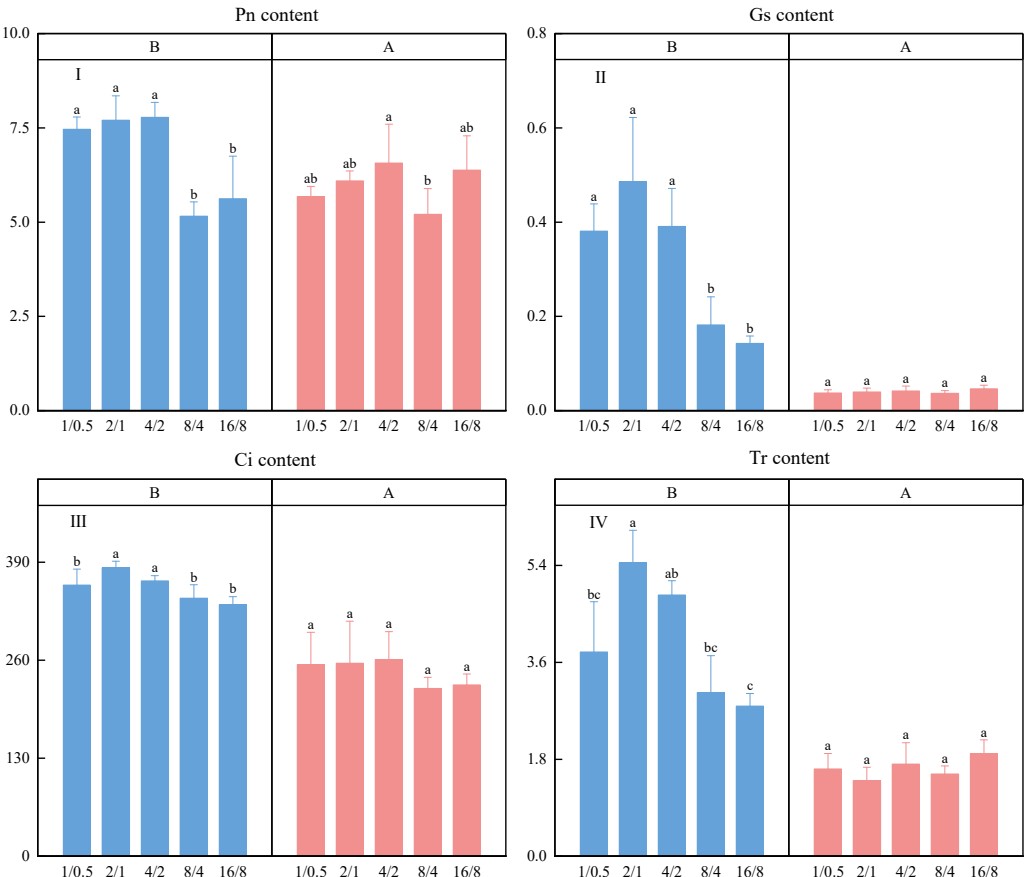

**Figure 4.** Bar plot of effect of different light supply modes on net photosynthesis rate (Pn), stomatal conductance (Gs), transpiration rate (Tr), and intercellular $CO_2$ concentration (Ci). There is a significant difference in the same parameter with different letters at the level of $p < 0.05$. (**I–IV**) Pn, Gs, Ci and Tr contents under intermittent light supply treatment before and after continuous light supply. B: Before continuous light supply; A: After continuous light supply.

3.6.1. Energy Allocation of Antenna Pigment Absorption [Y(II), Y(NPQ), Y(NO)]

Figure 5 shows the energy distribution absorbed by antenna pigments in different processing groups. From the percentage stacked bar chart, we can see that the energy absorbed by photosystem II in the 2/1 B treatment group was significantly higher than that in the other treatment groups before continuous light supply. Except for 8/4 B, the energy absorbed by PSII in all other treatment groups was higher than that in the control group by 16/8 B. After continuous light supply, the 2/1 A treatment group still had the highest energy absorption for photochemical reactions, but the 4/2 A treatment showed a decrease. However, the 16/8 A treatment significantly increased the energy absorption of photochemical reactions, indicating that the control group had good adaptability to light stress. When comparing the energy consumption of non-photochemical quenching, it was found that the 4/2 A and 8/4 A treatments showed varying degrees of increase, indicating that the light stress generated by continuous light supply had a negative impact on lettuce, causing the absorbed light energy to be used for heat dissipation.

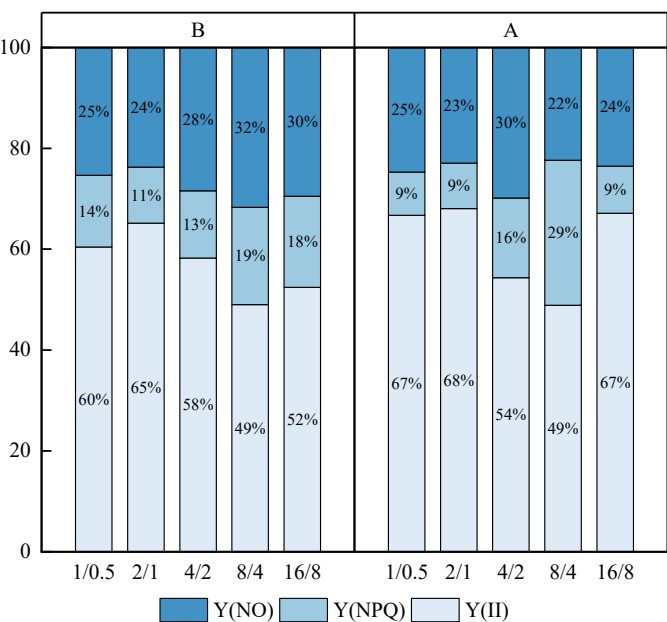

**Figure 5.** The percentage accumulation bar plot shows the effect of different light supply modes on Y(II), Y(NPQ) and Y(NO). B: Before continuous light supply; A: After continuous light supply.

3.6.2. Effects of Light Stress on Photosynthetic Apparatus ($F_v/F_m$, $\varphi_{P0}$, $PI_{ABS}$, $PI_{CS}$)

The dark adaptation value of Fv/Fm is usually used to evaluate plant stress. For leaves that are not under stress, the Fv/Fm value remains consistent, at approximately 0.83 [36]. As shown in Table 4, lettuce is in a stress state after intermittent and continuous light supply because light supply mode treatment causes the plant growth to break the inherent circadian rhythm. Through the fluctuations between different treatment groups, we can see that this stress behavior slows down after continuous light supply. $\varphi_{P0}$ can affect the performance indices $PI_{ABS}$ and $PI_{CS}$, thereby affecting the photosynthetic mechanism. The performance indices $PI_{ABS}$ and $PI_{CS}$ can more accurately reflect the state of plant photosynthetic mechanisms [36]. For $PI_{ABS}$, in Table 4, the $PI_{ABS}$ values of the 4/2 treatment were the highest before continuous light supply, and the $PI_{ABS}$ values of the 1/0.5 treatment were the highest after continuous light supply. Due to the significant fluctuations in $PI_{ABS}$ values between different treatments, $PI_{ABS}$ is more sensitive to light stress than Fv/Fm. $PI_{ABS}$ reflects the best photosynthetic performance under treatments of 4/2 B and 1/0.5 A. For $PI_{CS}$, both before and after continuous light supply, the $PI_{CS}$ treated with 1/0.5 is the highest.

**Table 4.** Effects of light stress on photosynthetic apparatus.

| Treatment | Fv/Fm | $\varphi_{P0}$ | $PI_{ABS}$ | $PI_{CS}$ |
|---|---|---|---|---|
| 1/0.5 B | 0.768 | 0.833 | 2.823 | 1502.909 |
| 1/0.5 A | 0.762 | 0.825 | 2.299 | 1533.408 |
| 2/1 B | 0.776 | 0.836 | 2.341 | 1055.216 |
| 2/1 A | 0.781 | 0.824 | 1.943 | 1497.348 |
| 4/2 B | 0.779 | 0.830 | 3.237 | 1333.866 |
| 4/2 A | 0.795 | 0.833 | 2.112 | 1472.879 |
| 8/4 B | 0.764 | 0.815 | 1.908 | 828.164 |
| 8/4 A | 0.791 | 0.829 | 1.270 | 1032.986 |
| 16/8 B | 0.771 | 0.821 | 2.186 | 438.015 |
| 16/8 A | 0.770 | 0.818 | 1.673 | 1091.067 |

1/0.5: Light period 1 h, dark period 0.5 h, repeated 16 times a day; 2/1: Light period 2 h, dark period 1 h, repeated 8 times a day; 4/2: Light period 4 h, dark period 2 h, repeated 4 times a day; 8/4: Light period 8 h dark period 4 h, repeated twice a day; 16/8: Light period 16 h, dark period 8 h. B: Before continuous light supply; A: After continuous light supply.

### 3.6.3. PSII Reaction Center Changes

In order to more accurately reflect the absorption, dissipation, transmission, and transformation of light energy by the photosynthetic organs of lettuce under different light treatments, this study measured and calculated the specific activity of the photosynthetic organs per unit area and the reaction center. Figure 6 describes the changes in the reaction center of PSII. Before continuous light supply, by comparing the numerical differences between different treatment groups, we can see that there are significant differences between the treatment groups in $ABS/RC$ and $ET_0/CS$. The intermittent light supply treatment had a significant impact on the PSII of lettuce, with the $ABS/RC$ of the 1/0.5 B treatment having the smallest and the most significant promoting effect on PSII. Through comparison, we can see that continuous light supply significantly reduced $ABS/RC$, $TR_0/RC$, $ET_0/RC$, and $ET_0/TR_0$, with no significant changes in $DI_0/CS$, while the quantum efficiency $ET_0/CS$ used for electron transfer was significantly improved, indicating that continuous light supply effectively improved the specific activity of the entire PSII photosynthetic mechanism. $ABS/RC = TR_0/RC \times ABS/TR_0$, and as shown in the figure, there is no significant change in $TR_0/ABS$, while $TR_0/RC$ decreases after continuous light supply, indicating a downward trend in $ABS/RC$. We know that ABS represents the energy absorbed by antenna pigments, RC represents the reaction center, and a decrease in $ABS/RC$ indicates an increase in the energy used for photochemical reactions, playing a positive role in the effective utilization of light from the continuous light supply. There is a connection between $TR_0/RC$, $ET_0/RC$, and $ET_0/TR_0$ ($ET_0/RC = TR_0/RC \times ET_0/RC = TR_0/RC \times \psi_0$). Based on the previous analysis, we know that $\psi_0$ decreases, and due to continuous light supply, the number of active reaction centers increases, resulting in a decrease in the $TR_0/RC$ value, itself resulting in a decrease in $ET_0/RC$.

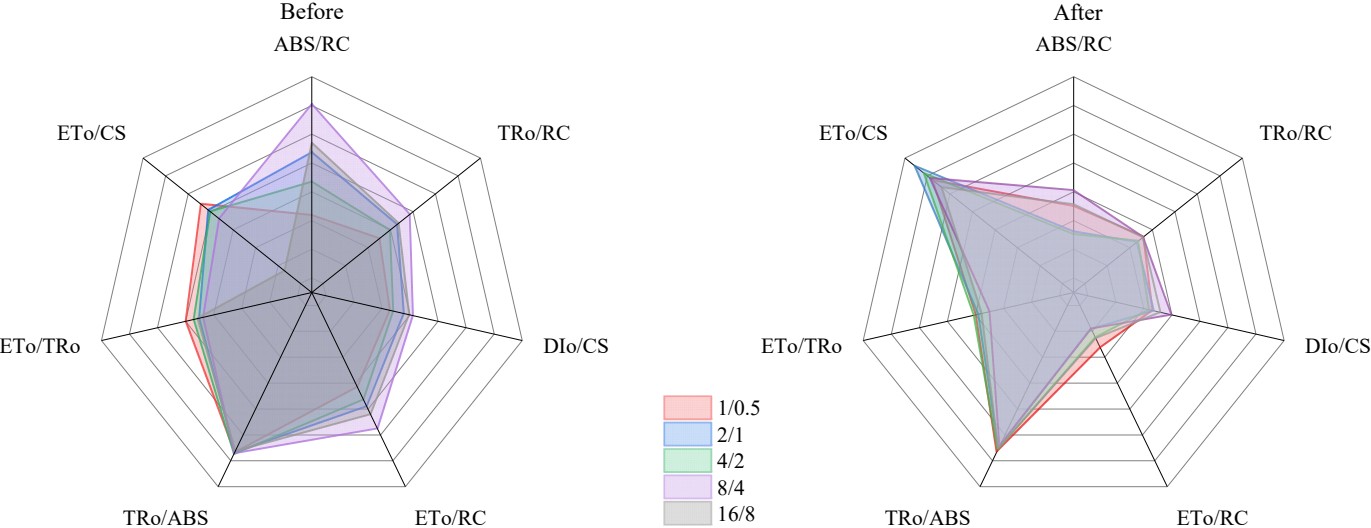

**Figure 6.** Spider diagram showing the change in the PSII reaction center.

## 4. Discussion

### 4.1. The Effect of Light Supply Modes on the Yield and Quality of Lettuce

The effects of light supply modes on lettuce growth, nutrient absorption, and photosynthesis were obtained through experiments. Through recording the growth indicators, we found that both before and after continuous light supply, intermittent light supply treatment created a better response in growth indicators than the control group (16/8), with the 2/1 treatment showing the most obvious response. Chen pointed out that intermittent light supply is more suitable for biomass growth under lower average light intensity [26]. Under intermittent light supply conditions, the lignin content in inedible biomass is lower, and intermittent light supply significantly affects the growth and physiological indicators of wheat [5]. The above experimental results are consistent with the results of this experiment.

It is worth mentioning that most experiments compare intermittent light supply and continuous light supply modes based on parallel experiments. Due to the fact that continuous light supply does not provide the dark period required for photosynthesis, continuous light supply is not ideal for producing a response in plant growth indicators. However, if only short-term continuous light supply is conducted before harvest, a better yield can be obtained on the basis of original growth. This is the basis of the idea of combining light supply modes in this article; the experimental results also confirmed this idea.

### 4.2. The Effect of Light Supply Mode on Nutrient Absorption

Chen et al. [37] pointed out that the single or combined spectra corresponding to the absorption peaks of chlorophyll a and chlorophyll b can enhance the root system's ability to absorb mineral elements Na, Fe, Mn, Cu, and Mo, with the single red spectrum having the most significant promoting effect. Lettuce grown under 20% blue light and 80% red light has the highest accumulation of Ca, Mg, Na, Fe, Mn, Zn, and B. Continuous pre-harvest light treatment significantly affects the accumulation of N, P, Fe, and Zn in hydroponic lettuce due to the interaction between LED light quality and nutrient liquid nitrogen forms [38]. Pre-harvest continuous light treatment increased the accumulation of C, N, P, Ca, and Mg in high nitrogen fertilizer and pre-harvest LED red blue light, but had no effect on the accumulation of trace elements [27]. In previous studies, light quality played a significant role in the absorption of nutrients in lettuce. This study indicates that light supply mode has a significant effect on nutrient absorption in lettuce. Shao pointed out that the accumulation of N, C, P, K, Ca, and Mg elements in lettuce was the highest when the light rhythm was 8 h and 4 h in the dark period [28], which is consistent with the conclusion in this study that the N, P, K, and Cu contents significantly increased after continuous light supply in the 4/2 and 8/4 treatments. The differences in nutrient elements Mg, Ca, Mn, and Zn between different treatment groups after continuous light supply were not significant, indicating that continuous light supply treatment can improve the problem of nutrient imbalance caused by intermittent light supply treatment in lettuce. There is no obvious pattern of Fe absorption in lettuce, which may be related to the element concentration of the nutrient solution. Continuous light supply causes stomatal closure, leading to a decrease in net photosynthetic rate. The reason for stomatal closure may be that light stress increases the permeability of the chloroplast membrane to abscisic acid, accelerates the transportation and accumulation of abscisic acid synthesized by the root system to the leaves, increases the content of abscisic acid in lettuce, and regulates stomatal closure.

### 4.3. Response Mechanism of Light Stress to Light Supply Mode in Lettuce

Under natural conditions, plants are affected by many adverse environmental stress factors [31], which can disrupt photosynthesis and lead to a decrease in plant productivity and total yield. Photosynthesis is particularly sensitive to environmental constraints [39], making photosynthesis measurement an important component of plant stress research. This paper aims to explain the response mechanism of chlorophyll fluorescence in lettuce under non-circadian rhythm light stress. Research on light supply modes shows responses that are different from the response of lettuce planted under conventional circadian rhythm. Under different light supply modes, lettuce shows a light stress effect due to the alternate cycling of light and dark periods that can be specifically shown in the Fv/Fm value of lettuce planted under light supply modes, which is less than 0.83 [40]. The changes in chlorophyll fluorescence in plants to some extent reflect the impact of environmental factors on plants. When the light energy absorbed by plants exceeds the requirements for photosynthesis, it will cause the inhibition of photosynthesis. The basic feature of this inhibition is the reduction of photosynthetic efficiency, or even light destruction [41].

When facing light stress, plants can avoid excessive absorption of light energy or adapt to it through self-regulation after being absorbed by plants. On the one hand, we can avoid the excessive absorption of light energy in plants by (a) changing the angle between

leaves to reduce excessive absorption of light energy; and (b) promotingthe dark movement of chloroplasts. Under strong light, chloroplasts aggregate on cell walls parallel to the direction of light to reduce light absorption [42,43]. Plants can also avoid excessive light absorption by (c) absorbing or filtering out strong visible and ultraviolet light through phenolic compounds such as anthocyanins, thereby reducing the intensity of the light reaching chloroplasts, thus reducing light inhibition.

On the other hand, the regulation of excessive light energy after absorption by plants is counteracted by the following mechanisms: (a) Dissipation in the form of thermal energy through non-photochemical quenching (NPQ) [44]. According to the different characteristics of photosynthetic induction and dark relaxation dynamics, NPQ is divided into three different components: high-energy state quenching (qE), light suppression quenching (qI), and state transition quenching (qT). High-energy state quenching (qE) refers to the dissipation of the ability to rely on the proton gradient across the thylakoid membrane. Its specific mechanism is as follows. When the assimilation force generated in the chloroplast exceeds the need for carbon assimilation, the thylakoid cavity acidifies, and the low pH induces the synthesis of zeaxanthin in the xanthophyll cycle and the protonation of PsbS protein [45]. These changes lead to conformational changes in the light harvesting pigment complex, resulting in the heat dissipation of excessive excitation energy. Light suppression quenching (qI) is the process of energy dissipation wherein the energy received by the photosynthetic mechanism exceeds the needs of photosynthesis, and photoinhibition occurs [46]. In this study, the energy distribution of lettuce after absorbing light energy was analyzed. Before continuous light supply, the NPQ of the treatment group was lower than that of the control group of 16/8. After continuous light supply, the energy ratio used for heat dissipation in all treatment groups increased, and the reason for this increase may be photoinhibition quenching (qI). (b) When PSII is overexcited, it is unable to produce enough NPQ, and the excess energy will produce superoxide, singlet oxygen, and hydrogen peroxide (ROS), thus damaging the oxygen-releasing complex of PSII [33,47]. Therefore, various antioxidants, such as anthocyanin [48], can eliminate this adverse effect. (c) Carotenoids can also play a photoprotective role by rapidly quenching the excited state of chlorophyll. During this study, it was observed that the content of carotenoids did not fluctuate significantly before and after continuous light supply. It may be that in this experiment, the photoprotective effect of carotenoids is not obvious. By analyzing the potential of light stress, we found that the adaptive response of lettuce to light supply modes may include all of the above situations. However, in this experiment, the main process coordinating this inhibition is light inhibition quenching (qI), which focuses on the adaptability of lettuce, reflected in chlorophyll fluorescence.

Continuous light supply throughout the entire plant growth cycle not only leads to light damage and disease [49,50], but also leads to economic issues such as high energy consumption and high cost in plant factories. Short-term continuous light supply can significantly improve growth indicators before lettuce harvest, thereby increasing yield [51]. Sacrificing some costs to obtain greater benefits may be a feasible option.

## 5. Conclusions

i. Intermittent light supply has significant effects on growth indicators, C, N, P, K, Mg, Fe, Ca, Mn, Zn, and Cu contents, chlorophyll a, chlorophyll b, carotene, photosynthetic indicators, and chlorophyll fluorescence indicators. The 2/1 treatment group had the best response in terms of lettuce growth, nutrient absorption and photosynthesis.

ii. After continuous light supply, the growth indicators of lettuce all significantly increased; the N, P, K, and Cu contents significantly increased in the 4/2 and 8/4 treatments. Continuous light supply also stabilized the Mg, Ca, and Mn contents between the treatment groups. Stomatal closure leads to a decrease in the net photosynthesis rate, transpiration rate, and intercellular $CO_2$ concentration. In summary, short-term continuous light supply can improve the yield and nutrient absorption of lettuce, but photosynthesis is hindered by stomatal closure.

iii.    Two light supply modes cause light stress in lettuce, which forces it to develop the ability to resist adversity, as evidenced by an increase in qI and $ET_0/CS$.

**Author Contributions:** Conceptualization, N.C., Y.L. and F.M.; methodology, N.C. and Y.Z.; software, N.C. and F.M.; validation, N.C.; formal analysis, N.C. and X.S.; investigation, N.C.; resources, Y.L. and Q.Y.; data curation, N.C. and X.S.; writing—original draft preparation, N.C.; writing—review and editing, N.C.; visualization, N.C. and Y.L.; supervision, Y.L., F.M., Y.Z., H.W. and Q.Y.; project administration, N.C. and Y.L.; funding acquisition, Q.Y. All authors have read and agreed to the published version of the manuscript.

**Funding:** This study was supported in part by the National Natural Science Foundation of China (51979134 and 51779113), Yunnan Fundamental Research (grant NO. 202101AT070125,202001AU070074), the Analysis and Testing Founding of Kunming University of Science and Technology (2022M20212214006), the Key Laboratory of Universities in Yunnan Province (KKPS201923009).

**Data Availability Statement:** The data are contained within the article.

**Acknowledgments:** We thank Qiliang Yang for their support and help in the research process.

**Conflicts of Interest:** The authors declare no conflict of interest.

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
