# Peer review of "The Mechanisms of Different Light Supply Patterns in the Nutrient Uptake and Chlorophyll Fluorescence of Hydroponic Lettuce"

_agronomy, doi:10.3390/agronomy13071910_

Round 1

Reviewer 1 Report

The authors tested the effect of intermittent and continuous light regimes on lettuce yield, nutrient accumulation, and photosynthetic efficiency. The abstract needs a concluding sentence. The introduction is informative about the light effects on plants. However, almost nothing is mentioned about the light effects on the nutrient uptake and chlorophyll fluorescence which is the manuscript’s subject. The authors also measured net photosynthesis and related parameters but it is mentioned in the introduction and the title. Moreover, the objectives are completely unclear and must be rewritten. In the materials and methods, the methodology is also not clear, and most of the methods were written in imperative form.

Specific comments are following.

L24-25. This is not necessary in the abstract.

L30. A concluding sentence is needed here.

L38. Please, explain the term LED and all abbreviations the first time they are mentioned.

L40. The term “supplementary” implies the presence of another light source, for example sunlight. In plant factories, artificial lighting is the only light source, thus it is not supplementary. This is also mentioned in L89.

L49. Not only red and blue. Other light ratios are also commonly studied.

L57-60. Is there a reference for this statement?

L86-87. I did not understand this statement. Did you perform a preliminary test?

L87-90. The hypothesis/objectives part is completely unclear. Please, rewrite this part because it is important for the reader to clearly understand the objective of the study even without reading the introduction.

L89. Please, do not start a sentence with “and”.

L101. Italics for Lactuca sativa.

L101-108. This whole paragraph uses imperative form and seems like a manual for sowing lettuce seeds. Please, rephrase. The same applies for L125-155, L164-182, L191-211.

L111. How much was the white light?

L118-120. Please elaborate on this since I am not sure I understood your methodology correctly.

L199. Here, you must mention details such as the CO2 flow rate and the conditions in the measuring chamber (CO2, temperature, PAR)

L218-220. Please, delete.

L326-333. Please review this and delete what is unnecessary.

L320. You may enlarge this figure to fit the page width and be easier to study. Moreover, the colours are faint, you may use a bit more intense colours like Figure 1.

L326-333. Please review this and delete what is unnecessary.

L367-371. Please review this and delete what is unnecessary.

L396-401. Please review this and delete what is unnecessary.

L431. A reference is needed for this statement.

L437-438. A reference is needed for this statement.

L498. Please insert the reference in the start of the sentence.

L582. What is this? Please, delete.

Moderate editing of English language required

Reviewer 2 Report

ABSTRACT:

The abstract must have a rationale, an objective, materials and methods, results, and conclusions.

The authors should mention the treatments and experimental design to explain the main findings.

Please insert the main results for the whole study.

Please Re-arrange the abstract as follows: The first sentence must be a rationale and please write a problem statement for this study. Then mention the objective, materials and methods, results, and conclusions. In addition, the abstract lacks clarity and specificity in describing the methods used to set up the intermittent light supply modes. The abstract mentions five intermittent light supply modes but does not provide details on how these modes were implemented or the specific light/dark periods used. Without this information, assessing the reliability and reproducibility of the study's findings is difficult. Additionally, the abstract does not mention any control group or compare the results of intermittent light supply modes to a continuous light supply mode, which limits the ability to draw meaningful conclusions about the effectiveness of the intermittent modes.

INTRODUCTION:

The introduction section is relatively short and missing main points such as role, importance and the effect of  intermittent light supply on growth, development, yield, and nutrient content of leafy crops such as lettuce, basil etc.

The introduction lacks a clear structure and flow. It jumps between different topics without proper transitions, making it difficult for readers to follow the logical progression of ideas. Reorganizing the content and adding clear subsection headings would improve the overall structure and readability of the introduction; however, there are some spaces for authors to enhance its quality further, which are as follows:

1. The introduction briefly mentions plant factories as efficient agricultural systems, but it lacks further elaboration on the concept, benefits, and significance of using plant factories for crop production. Providing more background information on intermittent light supply modes in plant factories would help readers understand the context and motivation for the study.

2. The introduction mentions that research on light intensity and quality has become more mature, but it fails to clearly articulate the existing gap in the literature regarding the in-depth study of light supply modes in the photoperiod. It is essential to highlight why studying light supply modes is necessary and how it addresses a research gap.

3. The introduction lacks a clear statement of the research objective. It should explicitly state what the study aims to investigate or achieve, such as understanding the effects of different light supply modes on lettuce growth, nutrient absorption, and photosynthetic system.

While the introduction briefly mentions some drawbacks of continuous and alternating light supply modes, it does not thoroughly discuss the limitations or potential negative effects associated with each mode. Providing a balanced discussion of the advantages and disadvantages of different light supply modes would enhance the comprehensiveness of the introduction.

Lines 61-71: The authors mentioned the effect of continuous light supply mode and continuous red light irradiation on tomatoes etc. But this study was focused on leafy crops such as lettuce. The authors should insert recent research about the effect of intermittent light supply modes in plant factories for lettuce or leafy crops, not fruity crops.  

Lines 86-96: Please remove this paragraph or shift to M &M section.

The last statement of the introduction should be the specific aims/objectives of this study.

2. Materials and Methods

The materials and methods subtitles should be re-arranged as the following:

1. Experimental site

2. Hydroponics setup and growing conditions.

3. Experimental design and treatments.

4. Nutrient solution preparation and management

5. Plant materials

6. Growth and yield of lettuce

7. Nutrients content of lettuce

8. Photosynthetic pigment measurement

9. Chlorophyll fluorescence parameters

9. Statistical analysis

However, the materials and methods are missing these points.

1.         The schematic diagram of the experimental setup of the experimental layout is not clear.

2.         The life cycle duration of each plant's is not mentioned. What about the planting density per square meter for lettuce?The age of seedlings when shifted to a hydroponics system

3.         Please include the following details:

The experiment started on day month year and ended on day month year. For example, The experiment started on 22 June 2019 and ended on 22 September 2019.

Line 100. Please correct the scientific name of lettuce to be italics as follows: (Lactuca sativa L.). What is the cultivar that used in this study?

4. Line 109 Experimental light treatment,

The way for writing light treatment is confused and the authors should write as treatment 1(T1, ect…).

What is the control treatment?

What is the experimental design in this study, and how many replications?

RESULTS AND DISCUSSION

The results and discussion section must be presented under specific subtitles.

The discussion section must be presented under certain subtitles, as the authors did for the results. This means as the authors presented their results under certain subtitles in Results, are they also suggesting developing subtitles under the Discussion section? The authors mostly only compare their results with the literature's results. However, they do not discuss the mechanisms by which the results are obtained.

5 Conclusion

The conclusion should have the main findings only.

References

Please check that the scientific name to be in italics, such as Reference 6.

The Quality of English Language in General good.

Reviewer 3 Report

The manuscript (agronomy-2465752) entitled “Mechanisms of different light supply patterns on nutrient uptake and chlorophyll fluorescence of hydroponic lettuce” present interesting topic and is quite well prepared. I found this paper to be need shorten and most objectively in write.

Beyond these quantifications, what is the broader outlook of your work? Reflect on whether your goals have been met, and consider the future implications of your findings. This should be encapsulated in your abstract.

Keywords in alphabetic order;

L39. Therefore,

L42. Remove “light”;

Check “far-red”;

Please use italic for Latin names of the species; for example Lactuca sativa (L101).

μmol m−2 s−1, don’t use “dot”

Figure 2. Not adequate;

Figure 3. Bar, not histogram;

Figure 7, need statistical comparison;

Need improve captions in tables and figures;

inconsistent reference style.

The conclusions are very general. Please add some more specific conclusions. In addition, note wite in topic.

Finally, I strongly suggest that the authors write the manuscript more succinctly and avoid verbose verbs and phrases. It is a bit confusing to interpret these data knowing about the subject; I think those encountering it for the first time will not understand what the authors want to explain.

I suggest that a native speaker assists the authors in rewriting in a simpler way, avoiding verbosity and prolix sentences.

Round 2

Reviewer 1 Report

The authors sufficiently addressed my comments and suggestions. Therefore, I suggest for the article to be published to Agronomy. Thank you for giving me the opportunity to review the manuscript.

Reviewer 3 Report

I thank the authors for addressing all my comments and doubts. I believe the authors have made extensive changes to the manuscript and have significantly improved it. Accept in present form.

Minor changes in grammar